# A Study on the Mechanisms of Nanoparticle-Stabilized High Internal Phase Emulsions Constructed by Cross-Linking Egg White Protein Isolate with Different Transglutaminase Concentrations

**DOI:** 10.3390/foods11121765

**Published:** 2022-06-15

**Authors:** Yanjie Zhao, Peng Wang, Yujuan Xu, Xianming Zeng, Xinglian Xu

**Affiliations:** Key Laboratory of Meat Processing and Quality Control, Ministry of Education, College of Food Science and Technology, Nanjing Agricultural University, Nanjing 210095, China; 2018108057@njau.edu.cn (Y.Z.); wpeng@njau.edu.cn (P.W.); 2018208021@njau.edu.cn (Y.X.); zxm@njau.edu.cn (X.Z.)

**Keywords:** egg white protein isolate, transglutaminase, high internal phase emulsion, nanoparticles, emulsion physical stability

## Abstract

There is an increasing interest in the development of high internal phase emulsions (HIPE) stabilized by food-grade nanoparticles due to their potential applications in the food industry. In this study, cross-linked egg white protein isolates (cEPIs) are prepared by adding 10 u/g, 20 u/g, and 40 u/g of transglutaminase (TG), and the impacts of interface properties of cEPIs and emulsifying of HIPEs are investigated. Relative to the native EPI, the cEPIs have more irregular and agglomerated morphology, and the turbidity and hydrophobicity are significantly increased. The particle size and zeta potential of cEPIs considerably varied with the addition of TG. In HIPE, the formation, physical properties, and microstructure are characterized by visual observations, the Turbiscan stability index, and CLSM. The results indicated that stable and gel-like HIPEs are formed by cEPIs at oil internal phase (*φ*) values of 0.75–0.90. Especially for the enzyme additions of 20 u/g, the cEPIs had the best storage stability and the lowest TSI value (2.50) and formed a gel network structure at *φ* values of 0.9 microscopically. Overall, this study can enrich the theoretical frame of interface properties by enzyme treatment. Besides, it would be of great importance for the research of HIPE stabilized by cEPIs appropriate to be applied in food formulations.

## 1. Introduction

HIPE is often referred to as a high concentration emulsion, which refers to the emulsion with a minimum oil phase concentration of 74% [1]. It is widely used in cosmetics, pharmaceutical preparations, tissue engineering, and petroleum fields. In the field of food, due to its high concentration of oil phase, vegetable oil can replace animal fat with fat replacement technology, which has received extensive attention [2]. However, in most of the HIPEs, the emulsifiers are mostly small-molecule surfactants, such as hydroxymethyl cellulose. Compared with general surfactants, the protein is derived from natural, non-toxic, and easy-to-extract sources, which can be accepted by the food industry, and it is favored due to its good surface activity and good emulsifying properties [3]. The egg white protein isolate (chicken protein isolate, EPI) selected in this experiment is a globular protein that is rich in disulfide bonds and non-polar amino acids, which endow the protein with better functional properties, such as foaming, gel properties, and interface adsorption properties [3,4]. However, due to their own limitations, EPI proteins are difficult to expose internal groups, so modifying the protein by modification methods can better help EPI to unfold its internal structure.

Enzyme modification of protein is an efficient way to improve protein functional properties, especially to improve protein thermal stability and salt stability [5]. A certain degree of hydrolysis can change the size of protein molecules, improve hydrophobicity, and can change the polarity of the molecule, the degree of ionization, and the enzymatic modification [6] can effectively improve the emulsification and foaming properties of the protein, and the cross-linking will affect the interfacial film thickness and layer density of the protein. One of the most commonly used enzyme-modified proteins is TG, which is produced by microbial fermentation. It can catalyze the formation of a covalent bond between the ε-amino group of lysine and the γ-hydroxyamide group of glutamic acid, resulting in the cross-linking of proteins [7]. As a cross-linking enzyme, TG can promote the unfolding of the internal structure of the protein. On the one hand, the hydrophobic group inside the protein is exposed, and on the other hand, the polymer can form a thicker protein film on the surface of the oil droplet and better stabilize the emulsion [8]. In previous studies, it had been proved that native EPIs could stabilize HIPEs at *φ* values of 0.75–0.80 [9], but with the higher *φ* values, due to the mutual extrusion of oil droplets, it is difficult for native EPIs to form a thicker protein film to stabilize the oil droplets, which leads to the aggregation and fusion of oil droplets.

As outlined above, the main objective of the present work was to investigate the potential of cEPIs as stabilizers to stabilize HIPEs with higher oil concentrations (*φ* values > 0.8). The influence of enzyme additions on the interface properties (including particle size, zeta potential, turbidity, surface hydrophobicity, and microstructure) of the cEPIs was characterized. The test properties of the HIPEs were also characterized in terms of visual observations, TSI, and CLSM.

## 2. Materials and Methods

### 2.1. Materials

EPI was purchased with a purity of >98% from Jinhai Food Industry Co., Ltd. (Qinhuangdao, China) and stored at −20 °C before use. The TG enzyme used in this study was purchased from Taizhou Dongsheng Food Technology Company (Taizhou, China), and soybean oil whose origin was Harbin, Heilongjiang Province, China, was purchased from Nanjing Suguo Supermarket (Nanjing, China), and the origin. All other chemicals were of analytical grade.

### 2.2. Preparation of Protein Suspensions

To prepare protein suspensions, EPI powder was dissolved in phosphate-buffered suspensions (PBS, 20 mM, pH 7) with a magnetic stirrer in order to prepare a 1% *w*/*w* protein suspensions (higher protein concentration would reduce the interface activity of the protein, and the ability to stabilize the interface would be reduced) and then hydrated overnight at 4 °C. Then, it was divided the suspensions into four groups. The first group was marked as S0 with no enzyme added (control group); the second group was treated with 10 u/g TG enzyme and marked as S1; the third group was treated with 20 u/g of TG enzyme and marked as S2; the fourth group was treated with 40 u/g of TG enzyme marked as S3 (Emulsions were strictly sealed and handled in a sterile room). After placing all the suspensions at room temperature to react for 24 h, the prepared protein suspensions were then stored at 4 °C for later use.

### 2.3. Preparation of the Emulsions

All the HIPEs stabilized by EPIs were obtained by homogenizing the mixtures of soy oil (*φ* values of 0.75, 0.80, 0.85, 0.90) and the EPI suspensions in the aqueous phase (1% *w*/*w*) for 60 s, using an Ultra Turrax T25 homogenizer (Precellys Evolution Super Homogenizer, Bertin Technologies, Montigny-le-Bretonneux, France) operating at 12,000 rpm.

### 2.4. Dynamic Light Scattering (DLS)

The particle size distribution of the native protein in the control group and the protein polymerization suspensions after cross-linking with different concentrations of TG enzyme was determined by DLS. After cross-linking, the complex dispersion and EPI suspensions were diluted 10 times with PBS (20 mM, pH 7.0) and then added to the sample tank in turn. The particle size was measured using a Zetasizer Nano-ZS 90 (Malvern Instruments Ltd., Worcestershire, UK) instrument equipped with a 4 mW He-Neon laser with a wavelength output of 633 nm at 24 °C and the monitoring angle set to 90 degrees.

### 2.5. Surface Hydrophobicity

To determine the surface hydrophobicity of EPIs and EPIs after cross-linking with different enzyme concentrations, a modified method was used from the literature [9]: 20 µL of a suspension of ammonium 8-anilino-1-naphthalenesulfonate (ANS) (20 mM in PBS, pH 7.0, 20 mM) was first added to 4 mL of the sample. After reacting for 20 min at room temperature (from 22 to 24 °C), the fluorescence spectrum of the sample was measured with a multi-plate reader. The excitation wavelength was 380 nm, and the emission wavelength was in the range of 410–570 nm. The same procedure for the control group was performed as well.

### 2.6. Turbidity

The protein turbidity after cross-linking of EPI and different enzyme concentrations was measured at 340 nm using a multifunctional microplate reader (SpectraMax M2, Molecular Devices Limited, San Jose, CA, USA), and the protein concentration was 1 mg/mL. All measurements were repeated 5 times.

### 2.7. Atomic Force Microscopy (AFM) Imaging and Analysis

The morphologies of the samples were compared by AFM (PeakForce Tapping Technology, Bruker Corporation, Karlsruhe, Germany) according to a previous report [10]. First, the samples were diluted to 0.05 mg/mL, and then a 5 µL sample was cast onto freshly cleaved mica. The sample was allowed to dry under ambient conditions for 20 min before being subjected to AFM analysis.

### 2.8. Storage Stability of the Emulsions

To test their storage stability, the emulsions with different oil phase concentrations after emulsification of freshly prepared EPI and TG enzyme cross-linked protein polymerization suspensions with different concentrations were immediately transferred to 25 mL transparent glass bottles after their creation. The emulsion changes of the samples after storage for 45 days were then observed at 25 °C to examine the storage stability of the different treatment groups under different oil phase concentrations.

### 2.9. Microstructure of the Emulsions

Each prepared 5 mL emulsion was stained with a dye that included 100 µL of 0.1% Nile red and 100 µL of 0.1% Nile blue. The dye and emulsion were vortexed to mix well and then incubated in the dark for 4–6 h before observation. We then transferred 10 µL of the staining emulsion to a microscope slide with a pipette and covered it with a coverslip. Next, we inverted the prepared sample on the sample stage and collected CLSM (confocal laser scanning microscope) images under a microscope using a 40× eyepiece at 25 °C [11]. Proteins and lipids can display red and green at excitation wavelengths of 633 nm and 480 nm [12], and each group of samples generated an image with a pixel of 1024 × 1024 through multi-point drawing, where a single point was collected multiple times to help ensure repeatable results.

### 2.10. Physical Stability of Emulsions

The physical stability of each emulsion was monitored by scanning a series of backscattered lights from the bottom to the top of each sample using Turbiscan multiple light scattering analyzer. The specific method is to transfer 25 mL of the emulsion into a cylindrical sample bottle, scan it at 24 °C for 3 h, and record the optical properties. The Turbiscan Stability Index (TSI) can be used to characterize unstable phenomena such as coalescence, flocculation, and emulsification, and is calculated by the data variation in the intensity of backscattered light at different locations of the sample cell, as given by the following equation:(1)$$\mathrm{TSI}=\sum i\frac{\sum h\left|scan_{i}-scan_{i-1}\right|}{H}$$

In the above formula, $scan_{i}$ represents the initial light intensity of the sample; $scan_{i-1}$ is the light intensity at the height of the sample at the last given time; *H* is the total height of the sample. The TSI sums all changes in the detection unit. Therefore, the relative stability of each system can be compared using the TSI, with a higher TSI value indicating a less stable system [13].

### 2.11. Statistical Analysis

All experiments were performed in triplicate, and the results were expressed as the mean ± standard deviation. The differences were significant at *p* < 0.05, which were determined by analysis of variance (ANOVA) using SAS 8.2 statistical software (SAS Institute Inc., Cary, NC, USA).

## 3. Results and Discussion

### 3.1. Particle Size and Zeta Potential

Particle properties are features that can affect basic protein properties [14]. As shown in Table 1, with the addition of the enzymes, the particle size of EPI increased from 463.10 nm to as high as 793.80 nm, indicating that the TG enzyme-catalyzed EPI formed larger or more polymers. Additionally, with the increase in enzyme addition, the particle size increased significantly (*p* < 0.05), which might be due to the further increase in particle crosslinking and the formation of more polymers [5]. Furthermore, the PDI (Polymer dispersity index) in Table 1 further showed that the EPI suspension system also tended to be more stable with the addition of enzymes, which meant that the precipitated flocs in the system were reduced.

Table 1 also showed that the zeta potential value of the group without the addition of the enzymes was −16.1 mV. In comparison, with the addition of the enzymes, the zeta potential value significantly increased. The increase in zeta potential might be explained by the fact that the TG opened the internal structure of the protein through amide action, which increased the negative charge on the surface of the protein, enhancing the electrostatic interaction and allowing more stable colloidal particles to form.

### 3.2. Surface Hydrophobicity

Surface hydrophobicity was an important indicator that could help to characterize the water-oil interface characteristics of protein particles in emulsions [15,16]. Generally speaking, the higher the hydrophobicity, the more lipophilic the protein particles were, and the more effectively they could become attached to the oil in the emulsion [2]. The ANS used in the present experiment was a hydrophobic molecular fluorescent probe, which was an important tool for studying the exposure of hydrophobic sites on the surface of proteins [17,18]. The surface hydrophobicity graph of EPI subjected to different enzyme additions is present in Figure 1.

Compared with S0, the surface hydrophobicity of cEPIs increased significantly. It was generally believed that enzymatic cross-linking caused proteins to form polymers either alone or with each other [12]. Under this framework, the addition of TG enzymes could result in the cross-linking of egg white separation, and this cross-linking could cause the EPI molecule to stretch and unfold, exposing the hydrophobic groups buried inside the EPI. The ANS had much more access to hydrophobic sites and could bind EPIs that were previously surrounded by a nonpolar environment, and thus the surface hydrophobicity increased. In addition, with an increase in the number of enzymes added, more EPIs could be cross-linked, which could lead to more EPI molecules unfolding and more hydrophobic groups becoming exposed, thus increasing the surface hydrophobicity.

### 3.3. Turbidity

Turbidity was an important indicator that reflected particle aggregations [19,20]. Compared with S0, the turbidity of cEPIs increased significantly after enzymatic cross-linking (*p* < 0.05) in Figure 2. Moreover, with an increase in enzyme concentrations, the turbidity of the S2 and S3 groups was significantly (*p* < 0.05) higher than that of the S1 group, which was consistent with the above particle size results.

From the results mentioned above, it might indicate that enzyme addition caused the EPIs aggregation and polymer formation, and it might further show that the TG changed the internal conformation of the protein. Combined with surface hydrophobicity, the cEPIs underwent the opening and aggregation of internal structure to form polymers, thus resulting in increased turbidity [15].

### 3.4. Changes in the Microstructure of EPI

AFM could reflect the microstructure of particles [21]. In order to further observe the particle microstructure morphology of different groups, the microstructure of the samples was analyzed by AFM [22]. The 3D and 2D images of the protein samples were collected in the field of view. Since the particles had a certain degree of swelling and water absorption, a certain degree of shrinkage also occurred during the drying process of the particles, resulting in a smaller average particle size [17,23]. Based on the height and width of the particles in the topography, it could be seen from the field of view of the atomic force microscope that the distribution of ovalbumin isolate particles in the control group was relatively uniform, the overall particles were smaller, and the brightness value was lower. However, with an increase in the enzyme concentrations, the degree of particle aggregation became more evident, the particle size of the particles gradually increased, and the height and brightness values of the particles also increased.

The height of the particles reached its maximum value at the enzyme concentration of 20 u/g, and many bright circles appeared, which indicated that the aggregation degree of the protein was better at this time and had exceeded the average particle size in Figure 3. This was consistent with the above particle size and PDI results and indicated that TG enzymes could better cross-link EPIs in the aqueous phase, promoting the unfolding of protein structures, the mutual attraction and repulsion between protein molecules, and the aggregation of protein particles, resulting in the formation of more and larger protein aggregates. Besides, it could also be seen from the visual field diagram that when TG concentration was 40 u/g (D: S3), the aggregation of protein particles was poorer than 20 u/g (C: S2), possibly due to excessive cross-linking.

### 3.5. Storage Stability of the Emulsions

Storage stability is an important indicator reflecting the physical stability of emulsions [24]. Figure 4 was the appearance drawing of the emulsion of each treatment group placed at room temperature for 45 days.

It was interesting to note that compared to the treatment group, the HIPEs at *φ* = 0.85 in the control group separated into two layers (scream layer and aqueous layer), indicating that the native EPI could not stabilize the emulsion. When the *φ* was 0.9 (the emulsion of the control group was not made successfully), serious demulsification occurred in the S1 group, while S2 and S3 only flocculated after 45 days in orange circle, suggesting that the emulsion was in an unstable state. Combining with the results from AFM and surface hydrophobicity, the cEPIs had higher hydrophobicity and polymerizability, thus having better lipophilicity. Furthermore, it was observed that the treatment groups had a thicker protein film in Figure 5, helping them to resist aggregation between oil droplets and providing them with better storage stability than the control group. As the *φ* was 0.9, due to the small amount of enzyme added in S1, the overall degree of polymerization was poor. Furthermore, combined with images from CLSM, it could be obviously seen that the protein film was so thin that it was difficult to fully wrap the droplet in S1.

### 3.6. Analysis of the Microstructure of the Emulsions

From Figure 5, it could be seen that in HIPEs with the *φ* = 0.75, the protein was distributed on the surface of the oil droplets and the continuous phase and that the oil droplets were similar to spheres. As the oil phase concentration increased, the oil droplets squeezed each other and deformed to form non-spherical shapes. Results of the previous studies have shown that the HIPEs gel network structure arose primarily because more proteins were adsorbed on the interface of oil droplets under the condition of a high oil phase ratio [25,26]. In Figure 5A, the radius of the oil droplet was larger, and the protein film could not better wrap the oil droplet at *φ* = 0.85; thus the aggregations would occur between the oil droplets, suggesting that the HIPEs were not in a stable state.

Additionally, due to lower hydrophobicity and weaker steric hindrance, the formation of the continuous protein film with the viscoelasticity was not easy [27,28], thus making it difficult for the protein to fully wrap oil droplets. However, the Figure 5 could be clearly seen closely packed and independent oil droplets and intact protein interface membrane at *φ* = 0.85 in treatment group, consistent with the results for storage stability. For the *φ* = 0.90, it could be seen from the Figure 5 that a thin protein film formed on the surface of the emulsions. This meant that with the addition of the TG, the protein membrane had a certain elasticity due to the hydrophobic interaction and hydrogen bonding of the treated proteins. Furthermore, the S2 formed a more complete protein film than S3 and S1 to stabilize oil droplets, which could explain why the HIPEs for S2 at *φ* = 0.90 still maintained a relatively stable state after 45 days of storage.

### 3.7. Physical Stability of the Emulsions

In order to comprehensively explore the stability of HIPEs with different concentrations, the TSI was used to characterize the short-term storage stability of different groups of emulsions within 3 h after their formations [23]. The TSI could document the dynamic instability of emulsions by summarizing changes in various destabilization conditions, including emulsification, flocculation, and coalescence [17,29]. Generally speaking, the higher the TSI value, the more unstable the system was [30,31].

The four graphs in Figure 6 show the TSI values for these different groups of *φ* values. The TSI values of all samples increased with storage time, indicating that the emulsions were always in a dynamically unstable state [23]. The control group showed a higher TSI value (5.70) at *φ* = 0.85 than the treatment group, indicating that the native EPIs were difficult to maintain a high concentration of oil phase in the absence of enzymes, which was consistent with the storage stability results. Combined with the results of particle size and surface hydrophobicity, this showed that after adding an enzyme, the cEPIs formed more aggregates and also developed a higher hydrophobicity, helping the cEPIs to adhere to the surface of the oil droplets and form a thicker and more elastic protein film. Furthermore, when the oil phase reached 90%, for S2, the TSI value was lower than for S1 and S3, suggesting that the emulsion system was more stable than S1 and S3. In the combination of the above indicators, the hydrophobicity and degree of polymerization of the protein were better, and a complete protein film could be formed on the surface of the oil droplet, so the physical stability of the HIPEs at this time was better than S1 and S3. Overall, with the addition of enzymes, the HIPEs had lower TSI values.

## 4. Conclusions

The current work provided important information about TG cross-linked EPI protein in a HIPE system. The addition of the TG enzyme induced the protein structure to unfold, and more hydrophobic amino acids from inside were exposed. Additionally, the addition of the TG enzyme helped the internal electrostatic charge to transfer to the surface of the protein molecule and also formed more and larger aggregates in the suspensions as compared to the control group. In the HIPEs, the enzymatically cross-linked proteins were able to stabilize oil droplets at an oil phase concentration above 85% due to their thicker protein films, while the control of EPI with no enzymes was not. This result provides a new direction for exploring the HIPEs stabilized by nanoparticles in the future.

## Figures and Tables

**Figure 1 foods-11-01765-f001:**
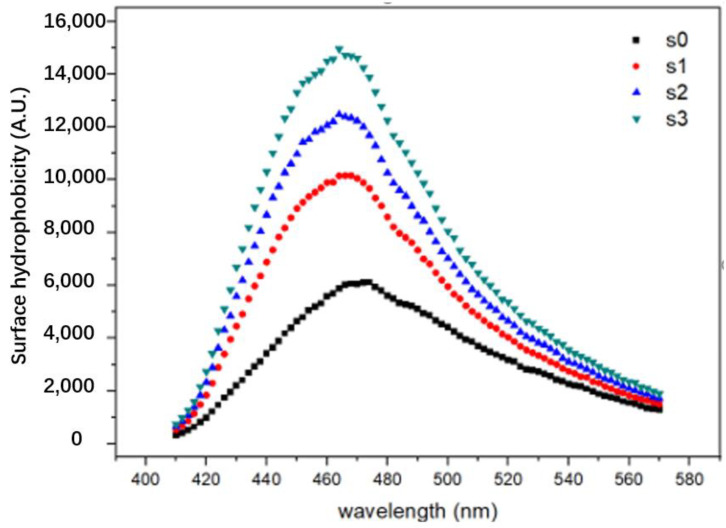
Fluorescence changes of EPI isolate cross-linked under different enzyme concentrations (0 u/g, 10 u/g, 20 u/g, 40 u/g).

**Figure 2 foods-11-01765-f002:**
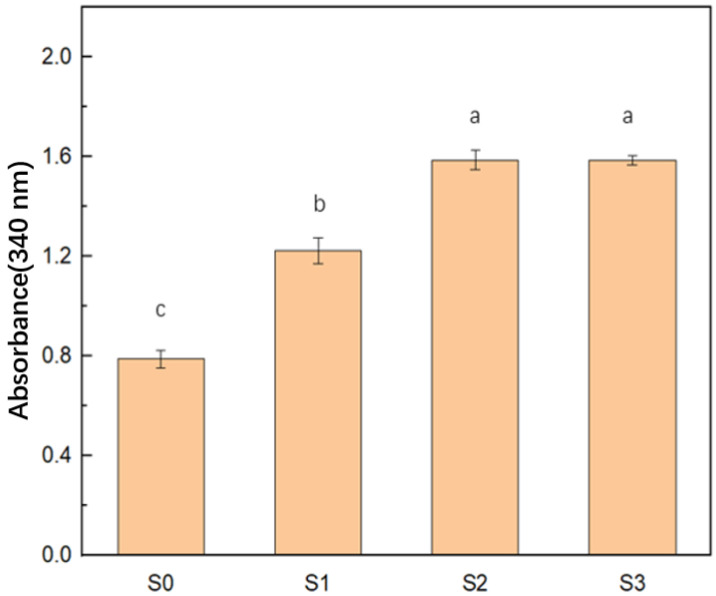
Turbidity value of EPI isolate cross-linked under different enzyme concentrations (0 u/g, 10 u/g, 20 u/g, 40 u/g). Different letters represent a significant difference at *p* < 0.05.

**Figure 3 foods-11-01765-f003:**
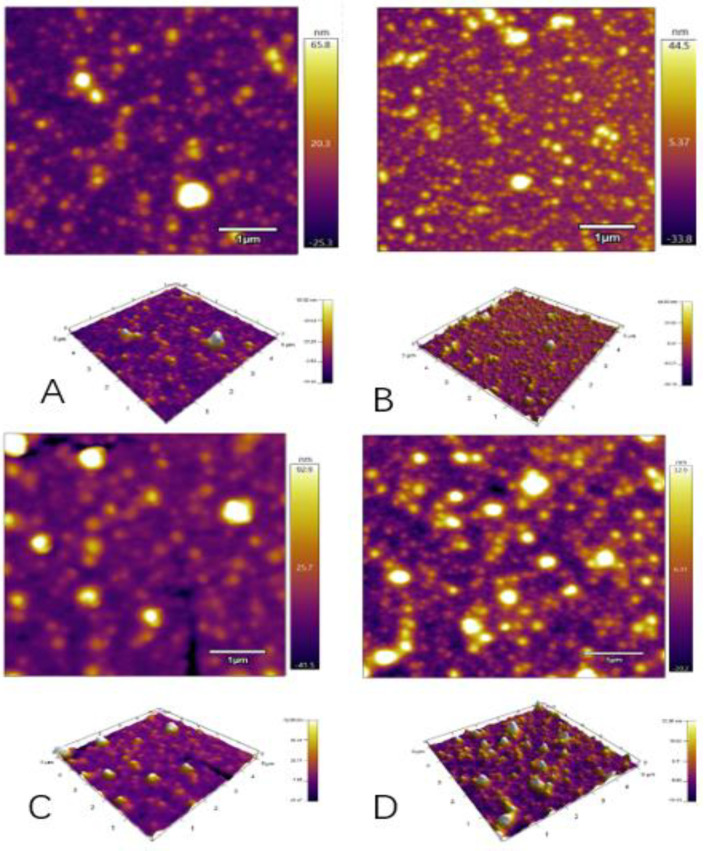
Atomic force microscopy images of EPI isolate cross-linked with different enzyme concentrations (**A**: S0, **B**: S1, **C**: S2, **D**: S3).

**Figure 4 foods-11-01765-f004:**
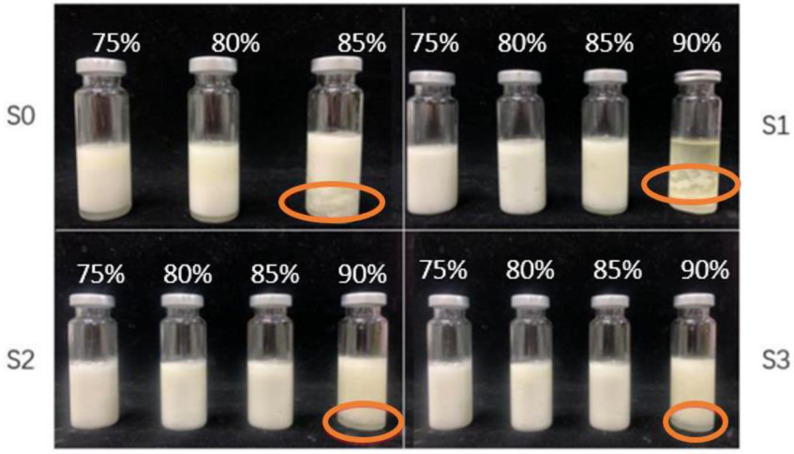
The appearance of the HIPEs prepared by EPI isolate cross-linked under different enzyme concentrations after 45 days at room temperature (25 °C). Flocculation phenomenon at orange circle mark.

**Figure 5 foods-11-01765-f005:**
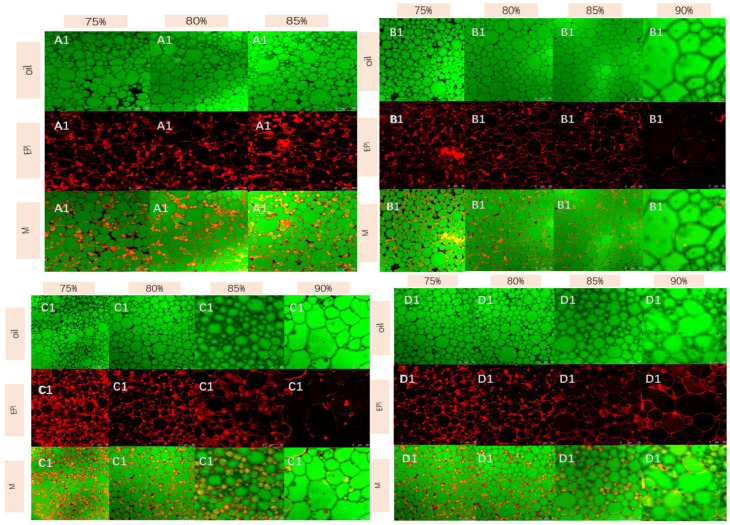
Confocal laser images (CLSM) of groups under different oil phase concentrations (75%, 80%, 85%, 90%) (**A1**: S0, **B1**: S1, **C1**: S2, **D1**: S3).

**Figure 6 foods-11-01765-f006:**
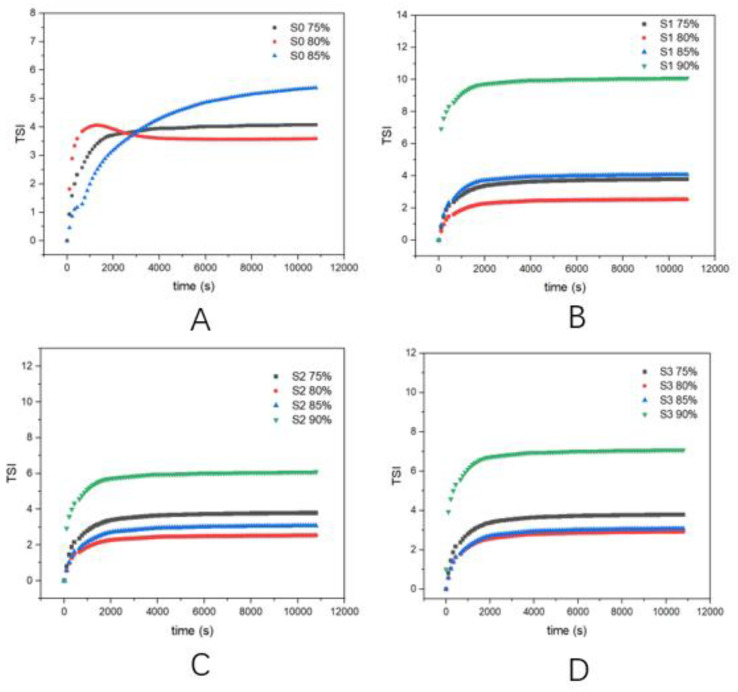
TSI (Turbiscan stability index) of groups under different oil phase concentrations (75%, 80%, 85%, 90%) as a function of time (25 °C). (**A**: S0, **B**: S1, **C**: S2, **D**: S3).

**Table 1 foods-11-01765-t001:** Average particle size, Zeta potential, and PDI value of EPI cross-linked under different enzyme concentrations (*n* = 5).

Enzyme Concentration	Particle Size (nm)	PDI	Zeta Potential (mV)
**S0**	463.10 ± 0.50 c	0.174 ± 0.000 c	−16.10 ± 4.24 b
**S1**	608.40 ± 6.54 b	0.598 ± 0.000 a	−18.00 ± 0.40 b
**S2**	856.30 ± 4.14 a	0.403 ± 0.000 b	−21.30 ± 0.54 a
**S3**	793.80 ± 5.24 a	0.326 ± 0.000 b	−22.60 ± 0.94 a

Data presented are means and standard deviations from five times. Different letters represent a significant difference at *p* < 0.05.

## Data Availability

Data is contained within the article.

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
