# Peer review of "A Study on the Mechanisms of Nanoparticle-Stabilized High Internal Phase Emulsions Constructed by Cross-Linking Egg White Protein Isolate with Different Transglutaminase Concentrations"

_foods, 2022, doi:10.3390/foods11121765_

Round 1
Reviewer 1 Report
This is an interesting topic related to the improvement of the mechanisms of nanoparticle-stabilized high internal phase emulsions constructed by cross-linking egg white protein isolate with different transglutaminase concentrations. The topic is very interesting, the research is of high quality and shows a comprehensive study described in a clear straightforward, and detailed form. Comments are shown below.
Highlights. The highlights are adequate.
Abstract. Please define acronyms (TSI, CLSM).
Introduction. This section provides clear and enough information about the research.
Material and methods.
- Line 85. Missing information from the homogenizer such as the city and country of origin.
- Line 91. Missing information from the Zetasizer Nano Zs 90 such as the city and country of origin.
- Line 105. Missing information on the model, brand, and the city and country of origin of the equipment used.
- Line 147. Missing information on the brand, city, and country of origin of the equipment used.
Results and discussion.
- The sentence of lines 292-293 is not understood
- The sentence of lines 307-308 is not understood, since fractions greater than 0.9 the TSI value are greater, which indicates greater instability.
- In line 312 the second unit has a superscript (-1)
- In line 341 separate the word when and start with a lowercase letter
Conclusions.
- The conclusions are adequate
Tables and figures.
- The Tables and figures are adequate
References. This section is adequate.
Reviewer 2 Report
foods-1705546
Title: A study on the mechanisms of nanoparticle-stabilized high internal phase emulsions constructed by cross-linking egg white protein isolate with different transglutaminase concentrations
The study describes a biomimetic approach to functionalize egg white protein by an enzymatic action. In particular, transglutaminase was utilized in various enzyme activities to induce protein-protein crosslinks which were subsequently used as stabilizer to generate emulsion with super high oil droplet concentrations. After carefully reading the manuscript, some major issues need to be addressed:
- What is the hypothesis of the paper?
- I believe that the crosslinking of proteins to generate aggregates could be supported by molecular weight measurements and is, as such, rather speculative. How was the enzyme activity of TG defined?
- The enzymatic treatment of a protein solution at room temperature for about 24 hours might cause some microbial issues. In addition, I believe that the emulsions formed and stored at room temperature for 45 days might also be microbiologically affected, which – in turn – influences the physical stability of the emulsion formed.
- The reader would benefit whether the emulsions stability would be defined as physical stability (e.g. changes in particle size distribution over time).
- In general, a protein concentration of 1% was used to prepare the emulsions at various oil droplet concentrations. The authors concluded that at high oil droplet concentration no stable emulsion could be formed. In case the protein concentration is increased, such as to fully cover the droplets surface, a stabilization could be possible. Can the authors comment on that issue?
- TSI measurements and long-term stability tests contradict each other.
- I believe that the crosslinking rate of TG under given conditions cannot be derived from the dynamic size measurements.
- In general, the figure and table captions need to be more precise. I would not recommend to use abbreviations.
Overall, the organization of the paper is somewhat misleading and could be shortened. In addition, a spell check is needed.
Reviewer 3 Report
Good paper. Some remarks are bellow.
Abstract: "cross-linking egg white protein isolates (cEPIs) were prepared" change on "cross-linked egg white protein isolates (cEPIs) were prepared".
2.11. Apparent viscosity of the emulsions...you could also look at thixotropic properties decreasing shear rates from 1000 to 0.1 s-1 (for future research)
Why you did not used thermal denaturared egg white proteins? Unlike native EWP, thermal denatured EWP are sufficiently cross-linked by MTGase. After explaining you can quote:
Farhad Alavi, Zahra Emam-Djomeh, Maryam Salami, Mehdi Mohammadian. Effect of microbial transglutaminase on the mechanical properties and microstructure of acid-induced gels and emulsion gels produced from thermal denatured egg white proteins. International Journal of Biological Macromolecules, 153, 2020, 523-532. doi.org/10.1016/j.ijbiomac.2020.03.008.
Round 2
Reviewer 2 Report
The authors should revise the manuscript based on the comments and suggestions addressed by the reviewers. The initial queries were not adequately addressed and incorporated.
